# Response of Phytoplankton Communities to Variation in Salinity in a Small Mediterranean Coastal Lagoon: Future Management and Foreseen Climate Change Consequences

Viviana Ligorini [1,2,*], Marie Garrido [3], Nathalie Malet [4], Louise Simon [1,2], Loriane Alonso [2], Romain Bastien [2], Antoine Aiello [2], Philippe Cecchi [5] and Vanina Pasqualini [1,2]

1    UMR 6134 SPE CNRS, Université de Corse, 20250 Corte, France; lsimon@engees.eu (L.S.); pasqualini_v@univ-corse.fr (V.P.)
2    UAR 3514 Stella Mare CNRS, Université de Corse, 20620 Biguglia, France; alonso_l@univ-corse.fr (L.A.); bastien_r@univ-corse.fr (R.B.); aiello_a@univ-corse.fr (A.A.)
3    Environmental Agency of Corsica, 20250 Corte, France; marie.garrido@oec.fr
4    Ifremer, Laboratoire Environnement Ressources Provence-Azur-Corse (LER/PAC), 20600 Bastia, France; nathalie.malet@ifremer.fr
5    MARBEC, Univ Montpellier, CNRS, Ifremer, IRD, 34095 Montpellier, France; philippe.cecchi@ird.fr
*    Correspondence: ligorini_v@univ-corse.fr

**Abstract:** Mediterranean coastal lagoons are particularly vulnerable to increasing direct anthropogenic threats and climate change. Understanding their potential responses to global and local changes is essential to develop management strategies adapted to these ecosystems. Salinity is a fundamental structuring factor for phytoplankton communities; however, its role under climate change is understudied. We hypothesized that salinity variations imposed by climate change and/or management actions could disturb Mediterranean lagoons' phytoplankton communities. To test our hypothesis, we performed two 5-day microcosm experiments in which natural phytoplankton assemblages from the Santa Giulia lagoon (Corsica Island) were subjected to three increasing (53–63–73) and decreasing (33–26–20) levels of salinity, to mimic strong evaporation and flash flooding, respectively. Results indicate that over-salinization inhibited growth and modified the assemblages' composition. Freshening, on the contrary, showed feeble effects, mainly boosting microphytoplankton abundance and depleting diversity at lowest salinity. In both experiments and under freshening in particular, initially rare species emerged, while photosynthetic activity was degraded by salinity increase only. We demonstrated that phytoplankton communities' structure and metabolism are strongly altered by the predicted implications of climate change. Such impacts have to be considered for future management of coastal lagoons (control of sea exchanges and watershed fluxes). This work constitutes a priority step towards the proactive adapted management and conservation of such as-yet-neglected ecosystems in the context of climate change.

**Keywords:** Mediterranean coastal lagoon; climate change; salinity; phytoplankton; photosynthetic activity; diversity

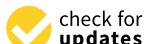



## 1. Introduction

The Mediterranean region is a hotspot of biodiversity and it also presents a millennial anthropogenic history. Its coasts host strong and constantly increasing urbanization and are hence subjected to unceasing anthropogenic pressures [1]. Moreover, it is universally considered a hotspot for future climate change in current projections [2,3].

Coastal lagoons are transitional systems characterized by strong physico-chemical gradients and an instable nature which make them particularly vulnerable to growing pressures [4,5]. The Mediterranean area is especially rich in coastal lagoons. Many of them are large lagoons, well studied and documented due to their usage and importance for

societal development and wellbeing, such as the Mar Menor in Spain, the Thau lagoon in France, and the Venice lagoon in Italy [6–9]. However, a multitude of small lagoons, i.e., those having a surface area smaller than 0.5 km$^2$, can be found along the Mediterranean coasts [7]. These small systems are understudied, as international protection norms and scientific surveys usually focus on larger lagoons. This is because these latter attract financial interest due to their uses and economic importance, but also because conservation norms usually apply a size limit for consideration, such as a minimum of 0.5 km$^2$ surface area for the European Water Framework Directive [10–12]. Small lagoons are especially threatened in the context of climate change since they have low mitigation potential due to their small surfaces and elevated perimeter/area ratio [9,13]. Nevertheless, these systems are very interesting to study due to their importance for ecosystems and human wellbeing but also because of their high reactivity, which makes them fragile regarding disturbance but also excellent sentinels for change [9,14,15]. In fact, since small lagoons have a high response reactivity, they act as "early signalers": understanding their reaction to change can provide useful knowledge about the potential response of larger lagoons in the future. Due to the vulnerability of these ecosystems, it is essential to understand their functioning in order to anticipate the potential responses of lagoon communities to current and forthcoming disturbances. This would help in promoting and guiding lagoon conservation and management decisions, and ultimately prevent the loss of the ecosystem services these systems provide.

Phytoplankton communities play a fundamental role in biogeochemical cycles and the functioning of lagoon ecosystems, where they are efficient consumers of nutrient inputs and fundamental elements for primary production and energy transfer. In general, phytoplankton communities react rapidly to any environmental changes and disturbances, and any modifications at the phytoplankton level can cause cascade effects on higher trophic levels [6,16–18]. Hence, phytoplankton is universally considered a relevant indicator for aquatic ecosystems' status and functioning [19]. The structure and dynamics of phytoplankton communities are influenced by multiple factors, mainly temperature, light, salinity, turbulence, and nutrient availability. Despite the common recognition of the role of salinity as a primary influence on phytoplankton community structuring, its effects under global climate change are still poorly understood compared to those of temperature and nutrients [20,21]. Some mesocosm or microcosm studies have already been performed to evaluate phytoplankton response to salinity variations, but they mostly regarded the marine environment or species [22] or brackish systems outside the Mediterranean region [23,24]. More frequently, studies of this kind have focused on temperature variations [25] or have been carried out in indoor laboratories [26]. Some studies in different aquatic environments have already revealed changes in phytoplankton communities linked to salinity variations: increased salinities often cause a loss in species richness and a decrease in biomass; however, published results about effects of salinity increase are disparate since abundance increase has also been observed under these conditions for brackish phytoplankton [23,26,27]. Also, freshening has been found to mostly impact growth rates and photosynthetic activity [28]. Nevertheless, the salinity tested remained in the range of the variation usually encountered in coastal ecosystems (i.e., 0.2–35). Under growing salinity gradients, changes in community composition seem to converge to the promotion of rare species, which resist stress and take the upper hand when dominant ones are inhibited [29].

Despite the fact that coastal assemblages and species adapted to transitional environments are considered more resistant to environmental changes and especially strong salinity variations, phytoplankton community changes have already been observed in highly instable transitional environments, like Intermittently Closed and Open Lakes and Lagoons (ICOLLs) [26,29]. Nevertheless, the responses detected are extremely variable, due to the local intrinsic variability of these systems, the salinity values at the beginning of experiments or observations, and their local specificities in terms of hydrological functioning, notably the degree of exchange with the marine environment [27,30–33].

Salinity gradients and fluctuations in coastal lagoons are driven by connection to the sea through sea inlets and freshwater inputs from their watersheds and rainfall. This

implies any artificial interventions, notably management actions on sea inlets and channels, which are commonly carried out in lagoon environment, can influence salinity variations. Furthermore, during summer, high temperatures generate strong evaporation, which results in increased salinity and sometimes the drying up of lagoons, depending on their connection to adjacent hydrosystems. With the progression of climate change, according to projections for the Mediterranean area, summer temperatures and drought are predicted to increase, thus exacerbating the intensity and frequency of these phenomena [2,3]. At the same time, projections also predict an increase in the magnitude and frequency of flash flood events, particularly during autumn [2]. These events provoke sudden massive freshwater inputs into lagoons, thus variating the salinity gradients in an abrupt way, and are already known to impact biotic communities in lagoon environments [34]. These events could thus become more and more frequent in the near future, subjecting lagoons to higher disturbance and frequent sudden salinity variations. Small Mediterranean lagoons are extremely vulnerable to these local and global changes, in relation to their location and small surface. There is a general need for proactive management strategies for their conservation, so it is particularly important to foresee potential responses of these systems to possible future scenarios, in order to measure impacts on biodiversity and ecosystem functions and to help the development of adapted management strategies in a changing context [35–37].

In this context, there is a need for knowledge about the potential effects of salinity variations linked to climate change, particularly in the lagoon environment. Thus, the main objective of this study is to evaluate experimentally the responses of lagoon phytoplankton communities to sudden salinity variations. For this purpose, we carried out two 5-day experiments, during which natural phytoplankton communities from the small coastal lagoon of Santa Giulia, in the south of Corsica (in the Mediterranean Sea), were exposed to strong sudden salinity variations. The chosen lagoon can be considered representative of small Mediterranean coastal lagoons and a good example as a sentinel of local and global change [9]. The two experiments were performed in different seasons in order to mimic potential future scenarios of global change. During the first experiment, performed in summer, the phytoplankton community was exposed to an increasing salinity gradient, representing potential drought conditions and salinization due to strong evaporation. During the second experiment, performed in autumn, the phytoplankton community was subjected to a decreasing salinity gradient, in order to mimic a flash flood event or in general sudden strong freshwater inputs from extreme rainfall. Three questions are addressed: (i) Will the lagoon phytoplankton community's structure and functioning be substantially impacted by the increasing and decreasing salinity stress? (ii) Will initial dominant taxa maintain their dominance under increasing salinity stress and what is the importance of the compensatory growth of rare species? (iii) What are the perspectives on the ecological functioning of these ecosystems under the predicted implications of climate change and what recommendations can be made for their future management?

## 2. Materials and Methods

### 2.1. Study Site

The Santa Giulia lagoon is a small shallow lagoon located on the south-eastern coast of Corsica Island (41°31′32″ N, 09°16′12″ E; max. depth = 1.5 m, mean depth = 0.3 m; Figure 1) and it extends for 0.23 km$^2$ on a north-south axis. Freshwater inputs are provided by a 15.5 km$^2$ watershed, with a few streams and groundwater from the associated alluvial aquifer, mainly at the southern end of the basin. Exchanges with the sea take place in the south-eastern part, through a small sea inlet and channel (Figure 1). The northern part of the basin is more confined, without direct exchange with the sea nor important freshwater inputs (Figure 1). The lagoon is property of the *Conservatoire du Littoral* (CdL) and managed by the Environmental Agency of Corsica (OEC) under its protection status (SAC as N2000 site, Habitats Directive). The lagoon is highly impacted by past and current human activities, and it is subjected to strong tourism pressure during summer. Recently,

its management encountered difficulties, as users of the beach often carry out unauthorized interventions to open the sea inlet when obstructed by sand and litter accumulation, in order to prevent unpleasant odors by maintaining water circulation. Isolation from the adjacent marine environment also caused a drying-up event in summer 2020 due to strong evaporation, with salinity values up to 126 [9]. Otherwise, from the little information and few data available in the grey literature, the lagoon is considered meso- to euryhaline [9,38]. The lagoon hosts small artisanal fishing activities and cattle farming takes place in the south-western area of its watershed.

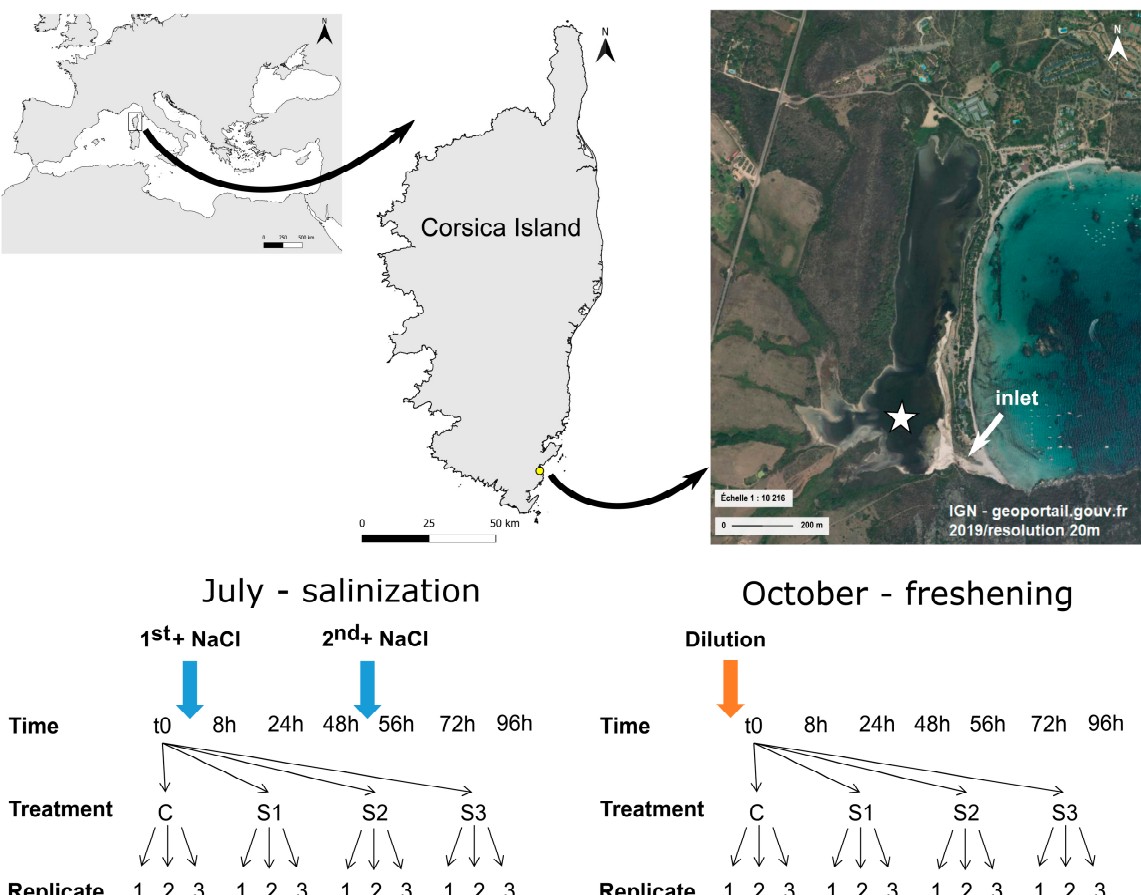

**Figure 1.** Sampling site location. Exact sampling point is represented by a white star. In the bottom panel, schematic description of experimental design is reported for both experiments (abbreviations used are the following: C for controls for both experiments, S1-S2-S3 for the three increasing or decreasing levels of salinity applied for salinization and freshening experiments, respectively).

The lagoon is subjected to typical Mediterranean climate, generally characterized by a three-month dry and hot summer season, from June to August, and two rainy periods, in spring and autumn. This configuration is evident from analyses of meteorological conditions over the standard reference period 1991–2020 established for France [9,39].

### 2.2. Microcosm Experiments

Two microcosm experiments were carried out in 2021 on phytoplankton communities collected in Santa Giulia lagoon. The first experiment involved the exposure of communities to an increasing salinity gradient and was conducted in summer, from the 7th to the 11th of July, while the second involved exposure to different freshening levels and was performed in autumn, from the 10th to the 14th of October. For both experiments, water samples were collected at the sampling station, located in the southern part of the basin (Figure 1). Meteorological conditions on the days preceding sampling were normal for both

seasons, with regard to the typical local Mediterranean climate [9], with 2 mm cumulative rainfall over the week preceding the October sample and no recent rain during the three days before sampling (14.20 mm cumulative rainfall since the 1st of September until the sampling day; Météo-France data (not shown)—La Chiappa weather station (41°35′41″ N, 9°21′47″ E). A 40 L volume of lagoon water was immediately filtered on a 200 μm mesh sieve in order to remove debris and mesograzers [22,25,40], and stocked in a polypropylene container for transport within two hours to the experimental location. In both seasons, the experimental setup was realized in an isolated 600 L plastic tank at the UAR Stella Mare laboratory (42°37′1.5″ N, 9°28′48.6″ E), installed outdoors in order to provide natural light and temperature fluctuations. The water temperature in the tank was monitored during all experiments through hourly measurements by HOBO® temperature probes. The tank used was designed to reflect natural temperature fluctuations thanks to isolating walls, so that effects due to potential higher temperature variations in the tank (linked to small size) compared to the natural system are excluded and fluctuations reflect natural conditions exactly.

Twelve microcosms were used for each experiment, with three replicates for each level of the treatment (S1, S2, S3) and control (C) (Figure 1). For the salinization experiment, we added salt in two successive steps, in order to minimize osmotic shock and to mimic a gradual evaporation. On the other hand, for the freshening experiment we applied the dilution in one step, in order to mimic a strong and sudden freshwater input, like in a flash flood event. The choice of the levels of the salinity treatment was based on recent observations at the study site and the few data available: in recent years, salinity in the Santa Giulia lagoon has ranged from 26.9 to 36.0 in autumn and from 33.8 to 42.2 in summer, but with peaks at >90 and up to 126 as well. These extremely high salinities were the result of the isolation of the lagoon from the sea due to the clogging of the inlet during the hot rainless season, which determined and exacerbated the evaporation phenomenon [9].

Each microcosm existed in a 3.5 L glass Schott-Duran bottle 80% filled with lagoon water (20% air), in order to maintain the systems close to their natural condition [40,41]. For both experiments, microcosms were nutrient-enriched and incubated around 20 cm deep in the tank, in order to mimic the sub-surface condition of the lagoon. Samples were collected at the beginning of the experiment (t0, just after filling and the addition of nutrients) and then at 8 h, 24 h, 48 h, 56 h, 72 h and 96 h from the beginning of incubation (Figure 1). The 8 h and 56 h intermediate samplings were chosen in order to evaluate potential responses over a short timespan (8 h) after the application of salinization stress [40] and then the same sampling design was kept for the two experiments. After each sampling, microcosms were randomly redistributed in the tank in order to avoid any bias linked to position. Salinity, temperature, and dissolved oxygen were measured at each sampling time with an YSI® ProDSS multiparameter water quality probe. Nutrient analyses were performed before the setup in the collected lagoon water, considered representative of the initial in situ water quality, and in the microcosms at t0, 48 h, and 96 h in 60 mL water samples filtered on Whatman GF/F filters (47 mm, 0.7 μm porosity) and further stored at −20 °C until analysis. Ammonium ($NH_4^+$) concentrations were determined through the fluorescence method [42], and nitrite ($NO_2^-$), nitrate ($NO_3^-$), dissolved inorganic phosphorus (DIP), and silicate concentrations were determined by the colorimetric method [43]. Dissolved inorganic nitrogen (DIN) was calculated as the sum of $NH_4^+$, $NO_2^-$ and $NO_3^-$.

### 2.2.1. Microcosm Setup for Salinization Experiment

For the summer experiment setup, each microcosm was filled with 2.8 L of filtered lagoon water (200 μm), immediately enriched by adding $NH_4Cl$. The choice of the form of N to be added was based on Leruste et al. [44] according to the seasonal processes driving nutrients fluctuations in Mediterranean lagoons [44–46]. On the other hand, the concentration of the selected nutrient form was calculated in order to double the initial theoretical concentration in situ for the season, thus obtaining non-limiting conditions and equivalent starting concentrations for all the microcosms. After t0, a first salinization was

applied by adding NaCl to the treatment microcosms, according to level (Table 1, Figure 1). Then, after 48 h, a second salinization was applied following the same procedure (Table 1, Figure 1). The initial salinity in situ was 43, so the salinization procedure resulted in final salinization of 53 for S1, 63 for S2, and 73 for S3 (Table 1).

**Table 1.** Description of experimental setup for the salinity levels of the two experiments: salinization, with two consecutive salt additions (after t0 and after 48 h samplings), and freshening, with one unique dilution at the beginning of the experiment. C stands for "Control" for both experiments, S1-S2-S3 indicate the three increasing or decreasing levels of salinity applied for the salinization and freshening experiments, respectively.

| Treatment | July-Salinization | | | | | October-Freshening | | |
|---|---|---|---|---|---|---|---|---|
| | Initial Salinity In Situ | First Salinization (% of Starting Salinity) | Salinity after First Salinization | Second Salinization (% of Starting Salinity) | Salinity after Second Salinization (Final Salinity) | Initial Salinity In Situ | Freshening (% of Starting Salinity) | Salinity after Freshening (Final Salinity) |
| C | | +0% | 43 | +0% | 43 | | −0% | 40 |
| S1 | 43 | +12% | 48 | +23% | 53 | 40 | −17% | 33 |
| S2 | | +23% | 53 | +46% | 63 | | −33% | 26 |
| S3 | | +35% | 58 | +69% | 73 | | −50% | 20 |

### 2.2.2. Microcosm Setup for Freshening Experiment

For the autumn experiment, each microcosm was filled with 1.4 L of filtered water lagoon (200 μm) plus a proportion of distilled water and/or lagoon water filtered at 0.2 μm, in order to obtain the right salinity level according to treatment but avoid different dilutions of the phytoplankton community and nutrient concentrations between treatments. Final volumes were 2.8 L. The initial salinity in situ was 40, hence the freshening procedure produced a final freshening of 33 for S1, 26 for S2, and 20 for S3 (Table 1). N enrichment was supplied in the form of $NH_4^+$ and $NO_3^-$, according to Leruste et al. [44] and seasonality, as previously explained, and $PO_4^{3-}$ was added too in order to avoid a strong N:P ratio unbalance. Nutriment additions were calculated in order to double the initial theoretical concentrations in situ for the season and thus obtain non-limiting conditions. Assuming a null nutrient input from the addition of distilled water, the final quantity added was calculated related to the initial volume of lagoon water present in each microcosm in order to obtain an equivalent final nutrient concentration between all microcosms at the start of the experiment, regardless of the treatment level.

### 2.3. Phytoplankton Community

The concentration of chlorophyll *a* (Chl *a*) was considered a proxy for phytoplankton biomass and measured at t0, 48 h, and 96 h. It was determined through spectrofluorometric analyses according to the method detailed in Neveux and Lantoine [47] on 100 mL water subsamples filtered on Whatman GF/F filters (25 mm, porosity 0.7 μm) and further stored at −20 °C until analyses.

At each sampling time, the community structure was investigated through flow cytometry analyses on 1.8 mL of fixed water samples (mix final concentration 0.25% glutaraldehyde and 0.01% pluronic (Poloxamer 188)), further stocked at −80 °C until analysis according to the method by Marie et al. [48]. Different small phytoplankton groups were distinguished based on their size and fluorescence: nanophytoplankton (NANO; >2 μm) and, among the picophytoplankton (<2 μm), autotrophic picoeukaryotes (PEUK) and picocyanobacteria (PCYAN), these latter obtained by cumulating the *Synechococcus*-like picocyanobacteria (*Synecho*-like) and *Prochlorococcus*-like picocyanobacteria (*Prochloro*-like) populations identified. Lastly, bacterioplankton was also taken into account, since it can play an important role in the structuring of phytoplankton communities through competition [49]. Thus, the heterotrophic bacteria population was also quantified through flow cytometry, following SyberGreen staining for green fluorescence.

In-depth community composition was assessed through pigment analysis performed by high pigment liquid chromatography (HPLC), at t0, 48 h, and 96 h, according to the method in Ras et al. [50]. To perform pigment extraction, water subsamples of 250 mL were filtered on Whatman GF/F filters (25 mm, porosity 0.7 μm) which were then stored at $-80$ °C until analysis. To investigate pigment composition, marker pigments were selected to represent major taxonomic groups (Table 2). The composition of phytoplankton communities was also determined at each sampling through a FluoroProbe® multi-wavelength fluorometer (BBE Moldaenke, GmbH, Schwentinental, Germany). The FluoroProbe® can discriminate four taxonomic groups: Bacillariophyceae/Dinophyceae, Cyanophyceae, Chlorophyta, and Cryptophyceae. Measurements were taken on 25 mL of dark-adapted subsamples and values were obtained as the mean of continuous measurement over 60 s. Values were then corrected according to Garrido et al. [51]. To better characterize the assemblages' changes, the microphytoplankton (>20 μm) community was investigated and quantified through counting techniques. Samples were taken in situ and at the end of the experiments. In order to determine the microphytoplankton community composition and abundances in the original natural environment at the starting point, 50 L of surface water was collected in situ and filtered through an Apstein plankton net (20 μm mesh). A 100 mL volume of concentrated sample was then fixed with formaldehyde at 2.5% final concentration. Then, in order to analyze the microphytoplankton community composition in the microcosms, at the end of the experiments (96 h), 1 L water samples were collected from each microcosm and fixed with Lugol solution (4% final concentration) and stored in dark refrigerated chamber until analysis. The use of different fixatives was decided in order to balance the advantages and disadvantages of each method, based on the availability of water volume, the conservation time desired, and the toxicity of the products. All samples were then examined independently according to the adapted Utermöhl method [52,53] with an inverted microscope (Olympus® CKX41, Olympus Corporation, Tokyo, Japan) at 40× magnification. At least 400 cells were counted (estimation error within ± 10% limits) [54,55] and identified at the most exclusive taxonomic level possible with verification according to several books [56–61] and databases (http://www.marinespecies.org/ (accessed on 12 July 2021), https://www.algaebase.org/ (accessed on 12 July 2021), databases available online). The Shannon's Diversity Index was calculated based on taxonomic units identified in the microphytoplankton communities. The use of different methods for sampling techniques between in situ and microcosm samples at 96 h does not prevent the qualitative comparison of results since the same counting strategy was applied, so a qualitative comparison of microphytoplankton communities was performed in order to describe the taxa present at the beginning at the end of the experiments [40,53].

**Table 2.** Marker pigments classification according to information seen in: 1. Claustre et al. [62]; 2. Vidussi et al. [63]; 3. Leruste et al. [64].

| Pigment | Taxonomic Group | Reference |
|---|---|---|
| Alloxanthin | Cryptophyta | 1, 2, 3 |
| Chlorophyll *b* | Chlorophyta and green flagellates | 1, 2, 3 |
| Divinyl Chlorophyll *a* | Prochlorophyta | 2 |
| Fucoxanthin | Bacillariophyceae | 1, 2, 3 |
| Lutein | Chlorophyta and Prasinophyta | 3 |
| Neoxanthin | Chlorophyta and Prasinophyta | 3 |
| Peridinin | Dinophyceae | 1, 2, 3 |
| Prasinoxanthin | Prasinophyta | 3 |
| Zeaxanthin | Cyanobacteria | 1, 2, 3 |
| 19′-Butanoyloxyfucoxanthin | Chrysophyta | 1 |
| 19′-Hexanoyloxyfucoxanthin | Prymnesiophyta | 1 |

Finally, metabolism was evaluated through photosynthetic activity efficiency measurements, performed with a Pulse-Amplitude-Modulated fluorimeter (Phyto-PAM Plankton Analyser; Heinz Walz GmbH, Effeltrich, Germany). Then, 30 mL water subsamples were

collected at each sampling time in opaque flasks and dark-adapted for at least 30 min before analysis. Samples were kept in a cooler and analyzed within 3 h from collection in order to respect recommended storage conditions for Phyto-PAM fluorometric analyses [65]. The ratio Fv/Fm, the maximum quantum yields of Photosystem II (PSII), was used as a proxy of the phytoplankton community's health [65] and it was calculated according to the following equation: Fv/Fm = (Fm − F0)/Fm, where Fm is the maximum fluorescence emitted under a saturating pulse of light (4000 µmol photons $m^{-2}$ $s^{-1}$) and F0 is the intrinsic initial fluorescence under non-actinic light after dark acclimation [66]. The effective quantum yield ($\Phi$PSII) was used as a proxy for the physiological state of the community and effective photosynthetic performance in the microcosms [67]. It was determined through the following formula: $\Phi$PSII = (F′m − F)/F′m, where F′m is the maximum fluorescence yield in light-acclimated state and F is the variable basal fluorescence under actinic light [66].

*2.4. Statistical Analysis*

All data analyses were performed through the statistical R software ([68]; version 4.0.2) and same analyses were performed for each experiment separately. Differences between levels of treatment and across time for different variables were investigated through mixed analyses of variance (mixed ANOVAs), including microcosms as a random effect in order to take into account temporal autocorrelation derived by repeated measures. When necessary, data were previously $\log_{10}(x + 1)$ or sqrt transformed to meet ANOVA conditions. In the case of the failed achievement of the normality assumption despite transformation, results were taken with caution and a more constraining level of significance ($p < 0.01$) was applied [69]. When significant effects of treatment, time or interaction were detected ($p < 0.05$ (or $p < 0.01$, if constrained)), pairwise post hoc tests were applied in order to examine differences. Changes in phytoplankton communities' composition were explored through HPLC pigment composition: reference pigments according to the literature [62–64] were kept and evolution over time for Controls was investigated through mixed ANOVAs. Afterwards, treatments were compared to Controls within each sampling time as percentage variations from the respective control. Finally, microphytoplankton total and class abundances, as well as Shannon's Diversity Indexes at 96 h, were analyzed through one-way ANOVA in order to compare differences in microphytoplankton total abundances, class abundances, and diversity between treatments at the end of the experiment. Significant effects of treatment ($p < 0.05$) were further investigated through Tukey's post hoc test.

## 3. Results

*3.1. Physico-Chemical Parameters*

*Salinization experiment*: The water temperature in the tank varied from 24.3 to 32.2 °C with a mean of 28.2 °C. In the microcosms, water temperature stayed in a range between a minimum of 23.6 °C and a maximum of 32.3 °C (mean water temperature: 26.7 °C). No significant differences between treatments were found; however, a significant time effect was detected (ANOVA $p < 0.001$, $F_{6,48} = 7850.49$, $\omega^2 = 1$; pairwise test: $p < 0.001$ for all contrasts). Salinity varied across treatments according to treatment levels imposed by experimental design (Table 1). Dissolved oxygen saturation percentage followed the temperature variations, and a significant effect of sampling time was detected (ANOVA $p < 0.001$, $F_{6,48} = 53.83$, $\omega^2 = 0.87$). Its values never fell below 83.8% with an overall mean of 96.2%. DIN and DIP concentrations were similar in all microcosms and no significant time or treatment effects were detected (overall mean [DIN] in µM: 7.61 ± 0.59 for C, 9.42 ± 1.55 for S1, 8.27 ± 0.39 for S2 and 8.40 ± 0.51 for S3; overall mean [DIP] in µM: 0.08 ± 0.03 for C, 0.13 ± 0.04 for S1, 0.10 ± 0.03 for S2, 0.13 ± 0.04 for S3). For silicates, again no difference between treatments could be detected (overall mean in µM: 88.62 ± 29.54 for C, 85.82 ± 28.61 for S1, 85.85 ± 28.62 for S2 and 84.43 ± 28.14 for S3) but a significant time effect was found (ANOVA $p = 0.025$, $F_{2,14} = 6.48$, $\omega^2 = 0.48$), indicating a higher concentration at 96 h compared to control (pairwise test: t0–96 h $p = 0.008$). At the end of the experiment, nutrients were still available, indicating no limitation occurred.

*Freshening experiment*: Water temperature in the tank followed circadian cycles and was in line with regional autumnal values. It never fell below 15.7 °C and averaged 17.9 °C, with a maximum at 19.9 °C. In the microcosms, no temperature differences between treatments were detected overall, while a significant time effect was identified (ANOVA $p < 0.001$, $F_{6,48} = 1222.25$, $\omega^2 = 0.99$). Salinity values were fixed for the different treatment levels according to the experimental design (Table 1). The percentage of oxygen saturation varied with time (ANOVA $p < 0.001$, $F_{6,48} = 92.74$, $\omega^2 = 0.92$), following the temperature pattern and never falling below 90.6%, with an overall mean at 96.3%. DIN concentration was similar between treatments but significantly varied over time (overall mean in µM: $1.53 \pm 0.65$ for C, $1.62 \pm 0.71$ for S1, $1.75 \pm 0.78$ for S2 and $1.79 \pm 0.78$ for S3; ANOVA $p < 0.001$, $F_{2,16} = 135.57$, $\omega^2 = 0.94$). DIP concentrations did not show any significant effect for treatment nor for time. However, silicate concentrations (overall mean in µM: $12.73 \pm 1.62$ for C, $12.60 \pm 2.38$ for S1, $10.12 \pm 1.28$ for S2 and $9.87 \pm 2.05$ for S3), varied significantly among treatments (ANOVA $p = 0.010$, $F_{3,8} = 7.63$, $\omega^2 = 0.74$) and sampling times (ANOVA $p < 0.001$, $F_{2,16} = 49.97$, $\omega^2 = 0.86$). A pairwise post hoc test indicated the highest values at 48 h from the beginning of the experiment (pairwise test: t0-96 h and 48 h–96 h contrast $p < 0.001$) and the lowest concentration at the lowest salinity treatment compared to the others (pairwise test: main contrasts C–S3 $p = 0.012$ and S1–S3 $p = 0.033$). Nutrients were still available at the end of the experiment, confirming that the experiment was performed under non-limiting conditions.

### 3.2. Phytoplankton Biomass and Small-Sized Phytoplankton Structure

*Salinization experiment*: Chl *a* concentrations varied from $0.77 \pm 0.07$ µg L$^{-1}$ at S3 at the beginning of the experiment to $1.37 \pm 0.24$ µg L$^{-1}$ in control at 96 h (Figure 2). Significant effects of both treatment (ANOVA $p = 0.003$, $F_{3,8} = 11.47$, $\omega^2 = 0.81$) and time (ANOVA $p = 0.008$, $F_{2,16} = 8.17$, $\omega^2 = 0.51$) were detected. An increase in time was identified (pairwise test: t0–96 h contrast $p = 0.019$; 48 h–96 h contrast $p = 0.052$), with an overall significant difference between the control and S3 treatments (pairwise test: C–S3 contrast $p = 0.003$).

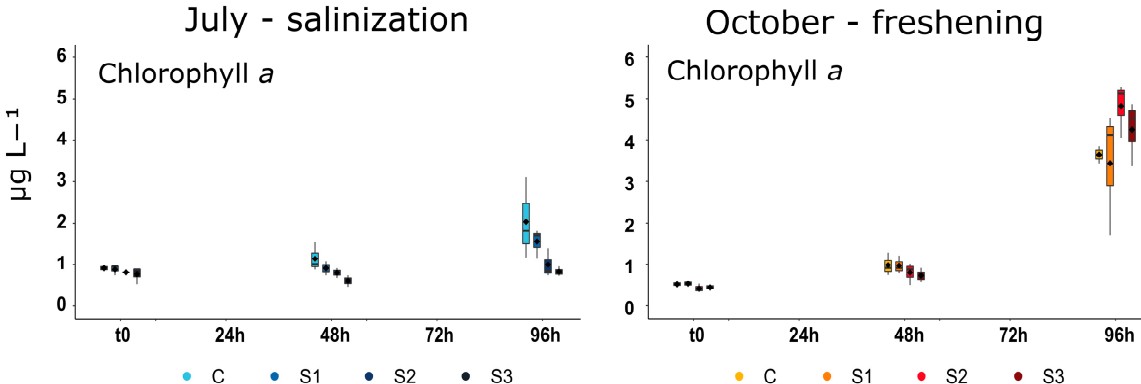

**Figure 2.** Biomass evolution over the experiments represented by concentrations of chlorophyll *a* pigment (µg L$^{-1}$). Black lozenges represent mean values. Color gradients indicate the levels of treatment applied, where C stands for "Control" for both experiments and S1-S2-S3 for the three levels of increasing salinity gradient in the salinization experiment and decreasing salinity gradient in the freshening experiment.

As for the small-sized phytoplankton structure, PCYAN densities increased only for Control, moving from $12{,}134.2 \pm 492.1$ cell mL$^{-1}$ at the beginning of the experiment to $14{,}619.4 \pm 6353.2$ cell mL$^{-1}$ at 96 h, while they decreased slightly for all salinization treatments (Figure 3). PCYAN was entirely constituted by the *Synecho*-like group. A significant effect of treatment and time interaction was identified (ANOVA $p < 0.001$, $F_{18,48} = 21.07$, $\omega^2 = 0.89$). The pairwise test confirmed a significant decrease for all salinization treatments and a significant difference between Control and S3 starting 24 h after the beginning of

the experiment. PEUK evolution over the experiment showed high variability and no clear pattern (Figure 3). The NANO group increased slightly for all treatments, up to $9996.3 \pm 3418.7$ cell mL$^{-1}$ in C at 96 h, except S3, which did not show any temporal evolution (Figure 3) but showed significantly lower values than other treatments, starting at 48 h (interaction: ANOVA $p = 0.008$, $F_{18,42} = 4.14$, $\omega^2 = 0.64$). On the other hand, heterotrophic bacteria showed a different trend (Figure 3): an effect of interaction was detected (ANOVA $p = 0.001$, $F_{18,48} = 6.50$, $\omega^2 = 0.71$), indicating that densities significantly decreased in all treatments except S3, which showed significantly higher values than C starting 24 h from the beginning, according to pairwise test results.

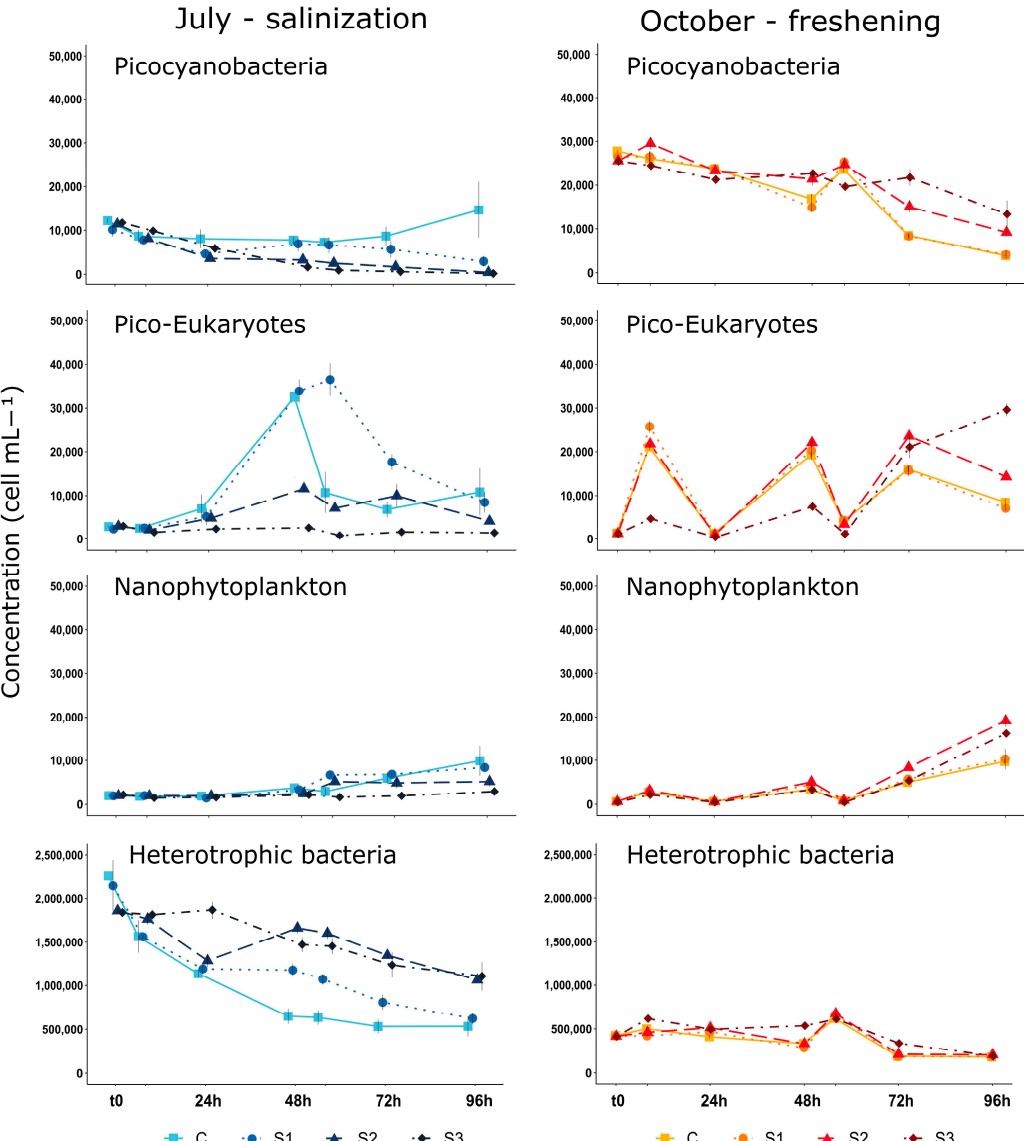

**Figure 3.** Evolution of pico- and nanophytoplankton and heterotrophic bacteria populations identified over the two experiments through flow cytometry analyses. Color gradients indicate the levels of treatment applied, where C stands for "Control" for both experiments and S1-S2-S3 for the three levels of increasing salinity gradient in the salinization experiment and decreasing salinity gradient in the freshening experiment.

*Freshening experiment*: Chl *a* concentrations significantly increased over time (ANOVA $p < 0.001$, $F_{2,16} = 220.65$, $\omega^2 = 0.97$; pairwise test: $p < 0.001$ for all contrasts), regardless of the treatment. They passed from an overall mean of $0.57 \pm 0.07$ µg L$^{-1}$ at the beginning of the experiment to a maximum of $4.82 \pm 0.22$ µg L$^{-1}$ in S2 at 96 h (Figure 2). The

structure of the small-sized community changed over the experiment and showed some differences between treatments. The PCYAN group was constituted of *Synecho*-like and *Prochloro*-like populations but was dominated by the former. A general significant loss of PCYAN over time was observed, moving from $27{,}862.9 \pm 291.2$ cell mL$^{-1}$ in the control at the beginning of the experiment to a minimum of $3912.5 \pm 104.0$ cell mL$^{-1}$ in the control at 96 h (Figure 3). The PEUK group experienced large fluctuations over the experiment. For both PCYAN and PEUK, a significant effect of interaction was found (PCYAN: ANOVA $p < 0.001$, $F_{18,48} = 14.54$, $\omega^2 = 0.85$; PEUK: ANOVA $p < 0.001$, $F_{18,48} = 81.89$, $\omega^2 = 0.97$), mainly indicating that C and S1 were significantly lower than S3 towards the end of the experiment, after 72 h (Figure 3). The NANO group increased slightly from 56 h from the start, and globally following the freshening gradient (Figure 3). A significant effect of interaction was detected (ANOVA $p = 0.004$, $F_{18,48} = 5.25$, $\omega^2 = 0.66$), indicating significantly higher values in S2 compared to C and S1 at the end of the experiment (pairwise test at 96 h: C$-$S2 contrast $p = 0.035$, S1-S2 contrast $p = 0.034$). The heterotrophic bacteria concentration ranged between $176{,}075.4 \pm 5977.6$ cell mL$^{-1}$ (C at 96 h) and $675{,}095.8 \pm 16{,}230.9$ cell mL$^{-1}$ (S2 at 56 h) and showed a general slight decrease (Figure 3).

### 3.3. Phytoplankton Communities' Composition by Pigment Analyses, Fluorometry, and Microscopy

*Salinization experiment*: ANOVA tests on Controls revealed the pigment concentrations did not vary significantly over time (Figure 4A), except for chlorophyll *b* (ANOVA $p = 0.009$, $F_{2,4} = 43.00$, $\omega^2 = 0.96$), 19′-butanoyloxyfucoxanhtin (ANOVA $p = 0.049$, $F_{2,4} = 7.00$, $\omega^2 = 0.78$), and alloxanthin (ANOVA $p = 0.014$, $F_{2,4} = 22.75$, $\omega^2 = 0.92$). Post hoc pairwise tests revealed a decreasing trend for chlorophyll *b* and alloxanthin, moving from $0.08 \pm 0.01$ mg m$^{-3}$ and $0.11 \pm 0.01$ mg m$^{-3}$ at t0, respectively, to $0.03 \pm 0.01$ mg m$^{-3}$ and $0.02 \pm 0.00$ mg m$^{-3}$ at 96 h, respectively, and a slight increase for 19′-butanoyloxyfucoxanhtin from below detection limit to $0.01 \pm 0.00$ mg m$^{-3}$ (Figure 4A). From comparison between treatments and corresponding controls, no differences in composition between treatments were detected at t0, i.e., the beginning of the experiment (Figure 4B). Afterwards, fucoxanthin showed systematically negative percentage deviances of treatments compared to C (S1: $-29\%$, S2: $-59\%$, S3: $-66\%$, at 96 h). Fluorometry data confirmed this pattern: the Bacillariophyceae/Dinophyceae group ranged between $0.56 \pm 0.03$ eq. µg Chl *a* L$^{-1}$ and $1.26 \pm 0.18$ eq. µg Chl *a* L$^{-1}$ in C at 8 h and 96 h, respectively, and significantly differed among treatments (ANOVA $p < 0.001$, $F_{3,8} = 20.03$, $\omega^2 = 0.88$), with Control values being higher than treatments, following the gradient (Figure 4C). Compared to C, 19′-butanoyloxyfucoxanhtin and 19′-hexanoyloxyfucoxanhtin were completely lost under salinization treatments, as well as peridinin (Figure 4B). A negative percentage deviation from C was detected for alloxanthin at 48 h (S1: $-36\%$, S2: $-55\%$, S3: $-64\%$) and 96 h ($-29\%$ for S1, S2 and S3) in all treatments, indicating a stronger decrease in Cryptophyta under salinization (Figure 4B). Nevertheless, no Cryptophyceae were detected by fluorometry (Figure 4C). A similar trend could be detected for cyanobacteria, represented by zeaxanthin, which showed a negative deviation from C of $-35\%$ and $-47\%$ in S2 and S3, respectively, at 96 h (Figure 4B). Regarding pigments related to green algae, at the end of the experiment, prasinoxanthin percentage deviations were positive for S1 ($+100\%$) and S2 ($+67\%$) but negative or S3 ($-67\%$), as well as neoxanthin (S1: $+100\%$, S2: $+50\%$, S3: $-50\%$) and lutein (S1: $+80\%$, S2: $+20\%$, S3: $-40\%$) (Figure 4B). According to fluorometry data, the Chlorophyta group showed an increasing tendency over time, except for S2 and S3, but also significantly lower values of S3 compared to C and S1 towards the end of the experiment, after 72 h (Figure 4C).



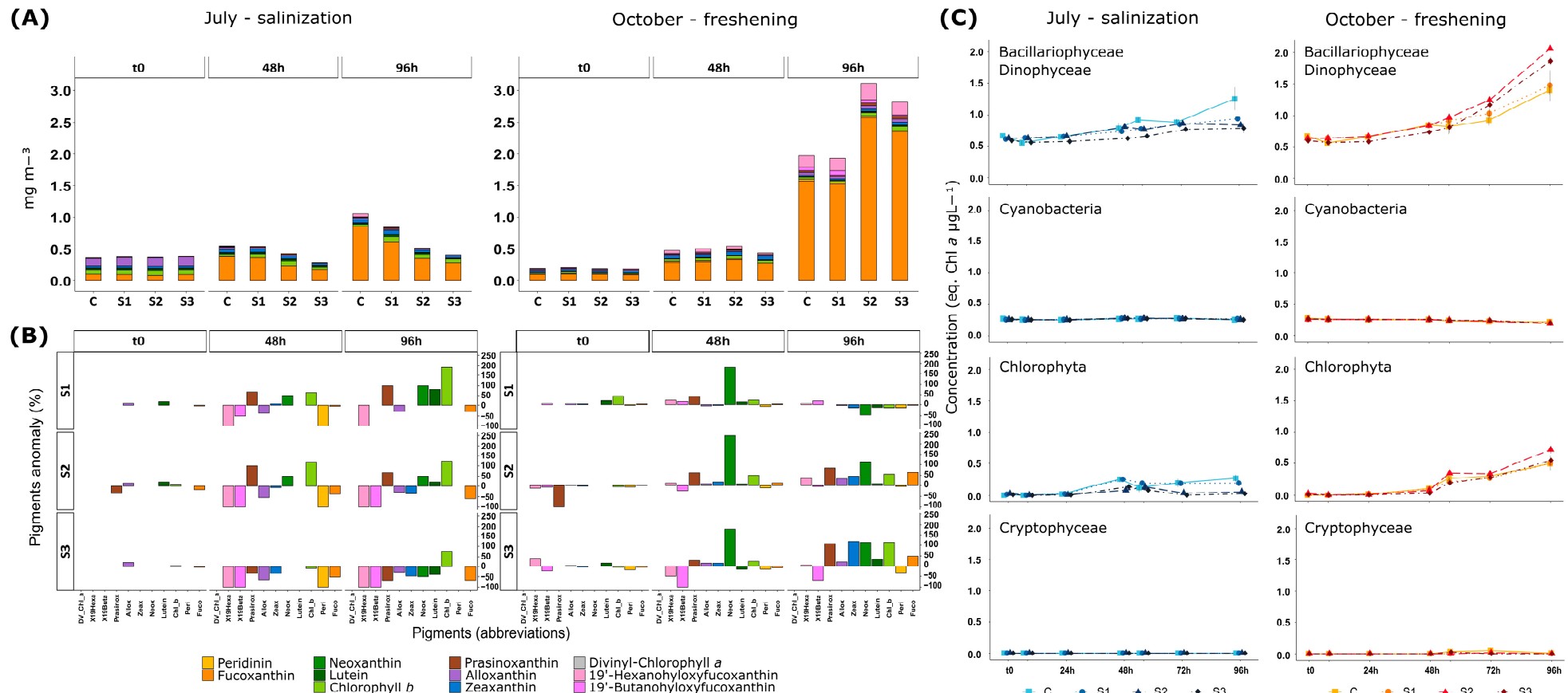

**Figure 4.** (**A**) Concentration of selected marker pigments quantified through HPLC analyses over the three sampling times; (**B**) evolution of community composition change expressed as percentage deviance of each pigment in each treatment to reference control at the same sampling time; (**C**) evolution of major phytoplankton groups identified by fluorometry for the two experiments. Color gradients indicate the levels of treatment applied, where C stands for "Control" for both experiments and S1-S2-S3 for the three levels of increasing salinity gradient in the salinization experiment and decreasing salinity gradient in the freshening experiment. Pigments are represented by different colors and abbreviations on the *x* axes (DV_Chl_a for Divinyl-Chlorophyll *a*; X19Hexa for 19′-Hexanohyloxyfucoxanthin; X19Buta for 19′-Butanohyloxyfucoxanthin; Prasinox for Prasinoxanthin; Allox for Alloxanthin; Zeax for Zeaxanthin; Neox for Neoxanthin; Chl_b for Chlorophyll *b*; Peri for Peridinin and Fuco for Fucoxanthin).

From microphytoplankton observations, the in situ community was dominated by diatoms and dinoflagellates, with *Navicula* sp. being dominant at $31 \times 10^3$ cell $L^{-1}$ (Figure 5; Table 3). Chlorophyta of the genus *Pyramimonas* were also present and reached $18 \times 10^3$ cell $L^{-1}$ (Table 3). At the end of the experiment, total density significantly differed among treatments (ANOVA $p < 0.001$, $F_{3,8} = 28.71$, $\omega^2 = 0.87$), indicating lower abundances following the increasing salinity gradient (Figure 5). This was mainly due to diatom growth, which also differed among treatments according to salinity gradient (Figure 5; ANOVA $p = 0.002$, $F_{3,8} = 13.41$, $\omega^2 = 0.76$), and to the loss of Chlorophyta, which only persisted in Controls (Figure 5; ANOVA $p = 0.001$, $F_{3,8} = 16.85$, $\omega^2 = 0.80$). Small undetermined cells, undetected in the field sample, largely developed during the experiment and especially in S1 and S2 low- and mid-salinity treatments (Figure 5; ANOVA $p < 0.001$, $F_{3,8} = 30.53$, $\omega^2 = 0.88$). Globally, *Navicula* sp., which was already dominant in the field, developed in all treatments, even if it reached lower densities at higher salinities (Table 3). Some species which were rarer at the beginning developed well in the microcosms, like *Nitzschia* sp., still following the salinity gradient (Table 3), but some other species were completely lost, like some Prorocentrales and mostly the Chlorophyta *Pyramimonas* sp., which only persisted in Controls at a mean density of $22 \times 10^3$ cell $L^{-1}$ (Figure 5; Table 3). Overall, Shannon's Diversity Index did not show any significant difference between treatments; however, a tendency could be detected, where the highest salinity treatment (S3) showed lower diversity than the others (Figure 5).

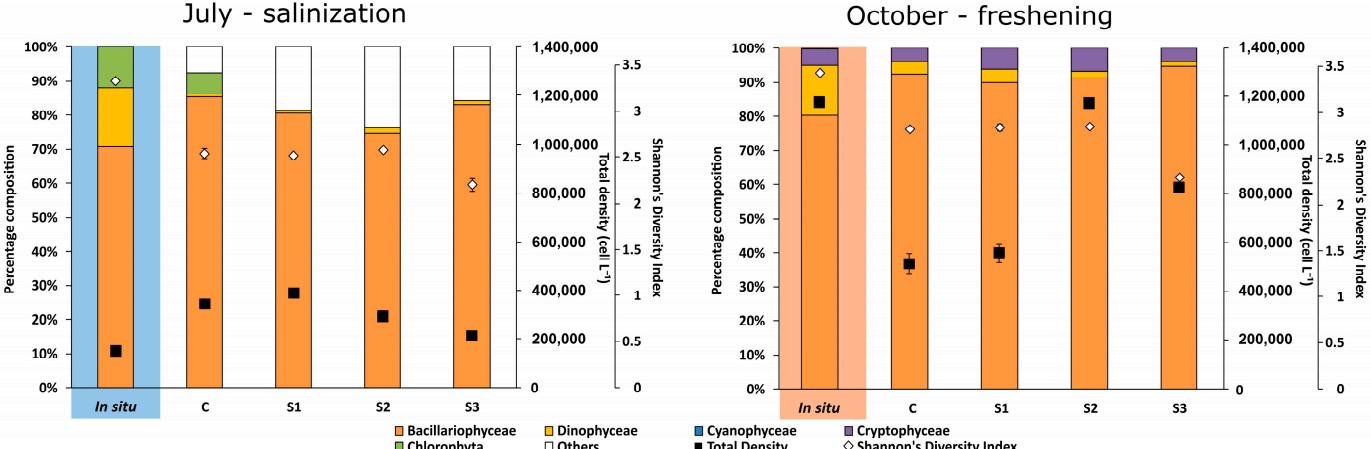

**Figure 5.** Community compositions in situ (left of each graph) and at 96 h obtained from optic microscopy observations. Composition is expressed as percentage contribution of major classes (scale on the left). Total cellular densities ± SD are also reported and represented by black squares (first scale on the right), as well as Shannon's Diversity Index ± SD, represented by white lozenges (second scale on the right). On the *x* axes, abbreviations correspond to control samples (C for both experiments) and the three different levels of treatment applied, i.e., S1-S2-S3 for the three increasing and decreasing salinity levels for the salinization and freshening experiments, respectively.

**Table 3.** Microphytoplankton composition of in situ and microcosm communities at the beginning and the end of the two experiments, respectively, expressed as cellular density for each species or taxonomic unit observed during microscopy analyses (mean values ± standard deviation over the three replicates). Taxonomic units are ranged in decreasing order of in situ abundance per each class, and color gradients represent magnitude of abundance in each column: higher values correspond to darker shades, hence to dominant taxa in the sample. C, S1, S2 and S3 correspond, respectively, to controls for both experiments and to the three increasing (salinization experiment) and decreasing (freshening experiment) salinity levels. The abbreviation "und." stands for "undetermined" and indicates that taxonomic classification could not be undertaken at more exclusive level.

**July 2021-Salinization (Cellular Abundance-Cell L$^{-1}$ ± SD)**

| Taxonomic Unit | In Situ | C | | S1 | | S2 | | S3 | |
|---|---|---|---|---|---|---|---|---|---|
| **Bacillariophyceae** | | | | | | | | | |
| *Navicula* spp. | 30,828 | 158,689 | ±35,288 | 163,999 | ±26,625 | 115,443 | ±22,675 | 105,124 | ±22,128 |
| Fragilariophycideae | 23,549 | 8259 | ±1842 | 5899 | ±1022 | 5870 | ±1752 | 2832 | ±811 |
| *Surirella* sp. | 12,845 | 0 | ±0 | 0 | ±0 | 0 | ±0 | 0 | ±0 |
| *Cocconeis* sp. | 11,132 | 9144 | ±2554 | 14,158 | ±9365 | 9494 | ±1957 | 10,088 | ±1405 |
| *Ardissonia* sp. | 9420 | 0 | ±0 | 0 | ±0 | 0 | ±0 | 0 | ±0 |
| *Nitzschia* sp. | 5994 | 61,057 | ±16,813 | 63,712 | ±19,467 | 42,618 | ±14,367 | 40,882 | ±8692 |
| *Petroneis* sp. | 3854 | 295 | ±511 | 0 | ±0 | 0 | ±0 | 0 | ±0 |
| *Licmophora* sp. | 3425 | 2655 | ±885 | 1770 | ±1770 | 0 | ±0 | 0 | ±0 |
| *Stauroneis* sp. | 3425 | 0 | ±0 | 0 | ±0 | 0 | ±0 | 0 | ±0 |
| *Cylindrotheca closterium* | 2141 | 15,633 | ±2703 | 5899 | ±2703 | 4059 | ±6436 | 0 | ±0 |
| *Diploneis* sp. | 1713 | 0 | ±0 | 0 | ±0 | 0 | ±0 | 0 | ±0 |
| *Entomoneis* sp. | 0 | 31,856 | ±20,809 | 47,194 | ±34,571 | 38,765 | ±8623 | 15,397 | ±10,737 |
| *Amphora* sp. | 0 | 5014 | ±2845 | 5309 | ±1770 | 2256 | ±915 | 2478 | ±1105 |
| *Chaetoceros* sp. | 0 | 2065 | ±3576 | 6489 | ±2044 | 239 | ±414 | 0 | ±0 |
| **Dinophyceae** | | | | | | | | | |
| Dinophyceae und. | 23,978 | 2360 | ±1022 | 2950 | ±2703 | 4887 | ±4559 | 2124 | ±1062 |
| *Mesoporos* sp. | 1713 | 0 | ±0 | 0 | ±0 | 0 | ±0 | 0 | ±0 |
| *Prorocentrum* spp. | 856 | 0 | ±0 | 0 | ±0 | 0 | ±0 | 0 | ±0 |
| *Dinophysis* sp. | 0 | 0 | ±0 | 0 | ±0 | 239 | ±414 | 0 | ±0 |
| **Chlorophyta** | | | | | | | | | |
| *Pyramimonas* sp. | 18,411 | 21,827 | ±9210 | 0 | ±0 | 0 | ±0 | 0 | ±0 |
| **Others** | | | | | | | | | |
| Small flagellates und. | 0 | 20,647 | ±2554 | 29,496 | ±5407 | 26,235 | ±6537 | 8672 | ±2146 |
| Small coccoid und. | 0 | 0 | ±0 | 42,474 | ±3540 | 40,671 | ±7797 | 23,715 | ±8648 |
| Raphidophyceae und. | 0 | 5899 | ±2703 | 0 | ±0 | 0 | ±0 | 0 | ±0 |
| und.1 | 0 | 0 | ±0 | 0 | ±0 | 2476 | ±1529 | 1239 | ±2146 |

**October 2021-Freshening (Cellular Abundance-Cell L$^{-1}$ ± SD)**

| Taxonomic Unit | In Situ | C | | S1 | | S2 | | S3 | |
|---|---|---|---|---|---|---|---|---|---|
| **Bacillariophyceae** | | | | | | | | | |
| *Licmophora* spp. | 233,609 | 5078 | ±3044 | 1327 | ±2299 | 20,352 | ±1533 | 7669 | ±2703 |
| *Nitzchia* spp. | 196,444 | 105,505 | ±12,743 | 120,698 | ±20,844 | 192,020 | ±19,925 | 129,193 | ±41,391 |
| *Navicula* spp. | 159,279 | 173,955 | ±67,309 | 135,033 | ±62,317 | 185,826 | ±23,143 | 163,409 | ±27,092 |
| Fragilariophycideae | 132,733 | 8502 | ±3054 | 6371 | ±1839 | 8849 | ±1533 | 0 | ±0 |
| Small centric | 69,021 | 48,206 | ±10,089 | 61,234 | ±9866 | 143,351 | ±14,047 | 34,216 | ±14,195 |
| *Petroneis* sp. | 47,784 | 0 | ±0 | 3451 | ±2871 | 0 | ±0 | 0 | ±0 |
| *Surirella* sp. | 37,165 | 0 | ±0 | 354 | ±613 | 0 | ±0 | 0 | ±0 |
| *Chaetoceros* sp. | 18,583 | 77,203 | ±22,813 | 101,585 | ±26,362 | 161,049 | ±12,545 | 87,309 | ±28,224 |
| *Cocconeis* sp. | 18,583 | 3853 | ±1263 | 3274 | ±668 | 5309 | ±4598 | 0 | ±0 |
| *Gyrosigma/Pleurosigma* sp. | 13,273 | 2008 | ±1991 | 442 | ±766 | 0 | ±0 | 0 | ±0 |
| *Diploneis* sp. | 7964 | 340 | ±589 | 0 | ±0 | 0 | ±0 | 0 | ±0 |
| *Amphora* sp. | 5309 | 0 | ±0 | 0 | ±0 | 0 | ±0 | 0 | ±0 |
| *Entomoneis* sp. | 2655 | 26,315 | ±7830 | 60,084 | ±11,201 | 324,752 | ±19,925 | 355,723 | ±54,863 |
| *Cerataulina* sp. | 0 | 2464 | ±2464 | 1770 | ±1533 | 17,698 | ±12,261 | 3540 | ±1770 |
| *Grammatophora* sp. | 0 | 18,760 | ±9047 | 5752 | ±9962 | 7964 | ±13,794 | 0 | ±0 |
| **Dinophyceae** | | | | | | | | | |
| Dinophyceae und. | 76,985 | 0 | ±0 | 0 | ±0 | 0 | ±0 | 0 | ±0 |
| *Alexandrium* sp. | 79,640 | 12,402 | ±807 | 13,539 | ±8281 | 17,698 | ±5526 | 2360 | ±1022 |
| *Prorocentrum micans* | 13,273 | 6643 | ±1655 | 1681 | ±853 | 3540 | ±1533 | 0 | ±0 |
| *Akashiwo sanguinea* | 0 | 442 | ±766 | 4513 | ±3326 | 2655 | ±2655 | 9439 | ±1022 |
| **Cryptophyceae Others** | 58,402 | 20,625 | ±1061 | 34,864 | ±7439 | 79,640 | ±5309 | 33,036 | ±5109 |
| **Dictyochophyceae** | | | | | | | | | |
| *Dictyocha* sp. | 2655 | 0 | ±0 | 0 | ±0 | 0 | ±0 | 0 | ±0 |

*Freshening experiment*: Analyses of phytoplankton composition indicated an evolution of the structure of communities over time and under different treatments (Figure 4A,B). ANOVA tests to detect potential evolutions of the pigments within the control level of treatment identified a significant growth over time of the following pigments: peridinin (ANOVA $p = 0.001$, $F_{2,4} = 182.33$, $\omega^2 = 0.99$), fucoxanthin (ANOVA $p = 0.001$, $F_{2,4} = 968.79$, $\omega^2 = 0.99$), 19′-butanohyloxyfucoxanthin (ANOVA $p < 0.001$, $F_{2,4} = 969.53$, $\omega^2 = 0.99$), 19′-hexanohyloxyfucoxanthin (ANOVA $p < 0.001$, $F_{2,4} = 1449.20$, $\omega^2 = 0.99$), alloxanthin (ANOVA $p = 0.017$, $F_{2,4} = 49.07$, $\omega^2 = 0.96$), and prasinoxanthin (ANOVA $p = 0.021$, $F_{2,4} = 30.18$, $\omega^2 = 0.94$). Parallel to this, a significant decrease in zeaxanthin was found (ANOVA $p = 0.010$, $F_{2,4} = 49.19$, $\omega^2 = 0.96$). Diatoms were the group contributing most to growth, since fucoxanthin concentrations passed from $0.10 \pm 0.00$ mg m$^{-3}$ to $1.58 \pm 0.04$ mg m$^{-3}$ at the beginning and after 96 h, respectively, in C, and reached up to $2.58 \pm 0.09$ mg m$^{-3}$ at 96 h in S2 (Figure 4A). Fucoxanthin percentage deviation from control at 96 h was at +64% and +49% for S2 and S3, respectively (Figure 4B). These results were confirmed by fluorometry data, where the Bacillariophyceae/Dinophyceae group largely dominated the community, reaching up to $2.06 \pm 0.01$ eq. µg Chl *a* L$^{-1}$ in S2 at 96 h, and showed a significant increasing tendency (Figure 4C). The significant effect of interaction (ANOVA $p = 0.007$, $F_{18,48} = 4.58$, $\omega^2 = 0.63$) highlighted a difference among treatments towards the end of the experiment, with C values being lower than those of other treatments starting from 72 h, according to pairwise test. The first level of treatment (S1) did not show particular pigments' percentage deviance from C, indicating a similar evolution of communities' compositions over the experiment (Figure 4B). On the other hand, S2 and S3 showed positive deviations from C at 96 h for most of the pigments, such as prasinoxanthin (S2: +85%, S3: +107%), alloxanthin (S2: +33%, S3: +20%), zeaxanthin (S2: +45%, S3: +118%), neoxanthin (S2 and S3: +113%), lutein (S2: +8%, S3: +33%), and chlorophyll *b* (S2: +55%, S3: +114%) (Figure 4B). Confirming the increase in pigments linked to green algae for S2 and S3, the fluorometry data highlighted significant higher values for the Chlorophyta group in S2 compared to other treatments (Treatment: ANOVA $p = 0.034$, $F_{3,8} = 4.80$, $\omega^2 = 0.64$; Figure 4C).

The microphytoplankton communities' quantification revealed some changes in the community composition and diversity from the starting community and among treatments at the end of the experiment (Figure 5). The in situ community was dominated by diatoms and dinoflagellates, and the Cryptophyceae class was also present with $58 \times 10^3$ cell L$^{-1}$ (Figure 5; Table 3). The starting community showed a high total density, reaching $1.2 \times 10^6$ cell L$^{-1}$, and a high diversity, with Shannon's Diversity Index at 3.42 (Figure 5). The community was mostly dominated by the diatoms *Licmophora* sp., *Navicula* sp. and *Nitzschia* sp. (Table 3). Among the dinoflagellates, *Alexandrium* sp. was dominant, reaching $80 \times 10^3$ cell L$^{-1}$ (Table 3). At 96 h, total abundances differed significantly between treatments (Figure 5; ANOVA $p < 0.001$, $F_{3,8} = 33.27$, $\omega^2 = 0.89$), with higher values following the decreasing salinity treatment gradient. This was mainly due to diatoms, which showed the same pattern as total abundances for differences among treatments (Figure 5; ANOVA $p < 0.001$, $F_{3,8} = 28.75$, $\omega^2 = 0.87$). Cryptophyceae developed at significantly higher values in low- and mid-salinity treatments compared to C and S3 (Figure 5; ANOVA $p < 0.001$, $F_{3,8} = 72.41$, $\omega^2 = 0.95$). Some species did not show much variation from the starting community and between treatments at the end of the experiment, like *Navicula* sp. and *Nitzschia* sp., which stayed dominant and developed well in all treatments (Table 3). Some species initially present in situ disappeared or were drastically reduced following the treatment, like most of the dinoflagellates (Table 3). Some other species that were initially extremely rare or not even detected proliferated during the experiment and were even boosted by the freshening treatment, for instance *Akashiwo sanguinea* and *Entomoneis* sp. This latter reached a mean of $325 \times 10^3$ cell L$^{-1}$ and $356 \times 10^3$ cell L$^{-1}$ in S2 and S3, respectively (Table 3). Globally, the strongest freshening treatment showed a significantly lower diversity compared to all other treatments (Figure 5; Shannon's Diversity Index: ANOVA $p < 0.001$, $F_{3,8} = 38.02$, $\omega^2 = 0.90$).

### 3.4. Phytoplankton Communities' Metabolism and Status

Salinization experiment: Fv/Fm ranged between $0.19 \pm 0.01$ in S3 at 8 h from the beginning and $0.51 \pm 0.01$ in C at 72 h (Figure 6), and significantly differed between treatments and sampling times (treatment: ANOVA $p = 0.006$, $F_{3,8} = 9.10$, $\omega^2 = 0.77$; time: ANOVA $p < 0.001$, $F_{6,48} = 141.41$, $\omega^2 = 0.95$), indicating a global increase over time opposite to the salinity increase, according to a pairwise test. The $\Phi$PSII variable experienced higher variability and ranged between $0.17 \pm 0.01$ in S2 and S3 at 8 h and $0.34 \pm 0.00$ in C at 56 h (Figure 6). A significant effect of the interaction was detected (ANOVA $p = 0.009$, $F_{6,42} = 3.89$, $\omega^2 = 0.63$), and a pairwise test confirmed the general tendency towards a slight increase, except for S3. Also, starting 24 h after the beginning of the experiment, C systematically showed significantly higher values than the other treatments.

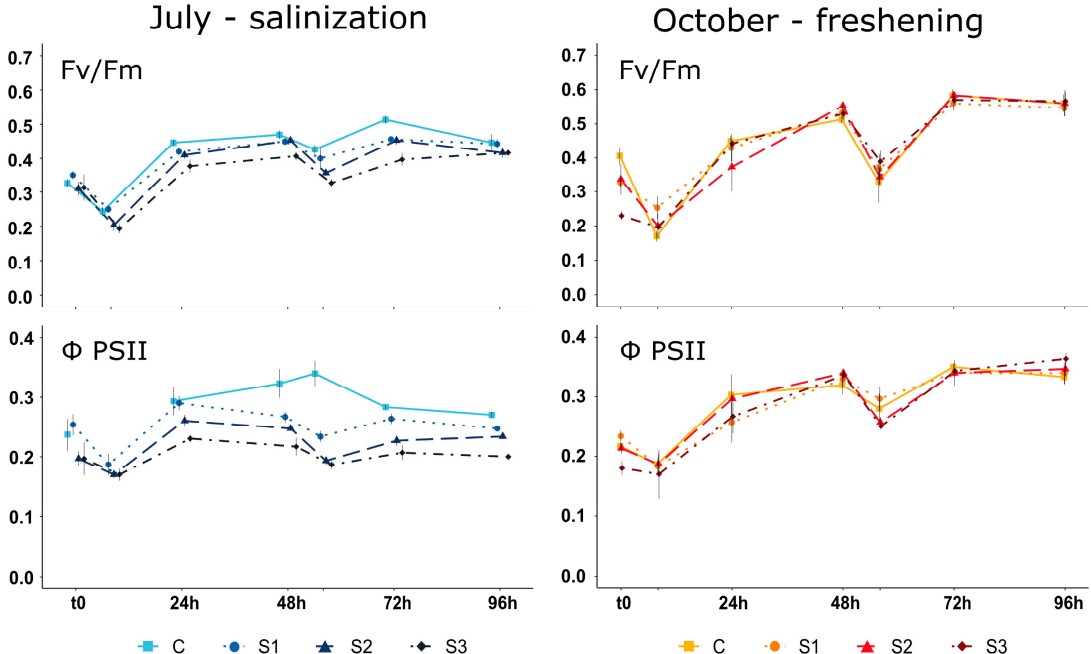

**Figure 6.** Evolution of photosynthetic activity's proxies Fv/Fm (top) and $\Phi$PSII (bottom) over the two experiments. Color gradients and line types indicate the levels of treatment applied, where C stands for "Control" for both experiments and S1-S2-S3 for the three levels of increasing salinity gradient in the salinization experiment and decreasing salinity gradient in the freshening experiment.

Freshening experiment: Fv/Fm ranged between $0.17 \pm 0.02$ in C at 8 h from the beginning and $0.58 \pm 0.02$ in C and S2 at 72 h, while $\Phi$PSII varied from $0.17 \pm 0.04$ in S3 at 8 h to $0.36 \pm 0.01$ in S3 at 96 h (Figure 6). No significant difference between treatments was detected, but a significant effect of sampling time was found for both variables (Fv/Fm: ANOVA $p < 0.001$, $F_{6,48} = 80.24$, $\omega^2 = 0.91$; $\Phi$PSII: ANOVA $p < 0.001$, $F_{6,48} = 57.09$, $\omega^2 = 0.88$), which indicated a global increase over time, according to pairwise comparisons.

## 4. Discussion

### 4.1. Global Effects of Short-Term Salinity Changes

Our two experiments clearly revealed that sudden changes in salinity altered the phytoplankton communities of the small Mediterranean coastal lagoon being studied. For both the experiments, based on the consideration of 8 h and 56 h samples, no significant short-term effect after stress application emerged, except for photosynthetic efficiency. Nevertheless, as lower photosynthetic efficiency was detected at 8 h and 56 h sampling times for both experiments, the depleting effect does not seem attributable to salinity stress, but rather to the time of the sampling. In fact, both 8 h and 56 h samplings were performed in the afternoon, while all the other measures were taken in the early morning,

so a reduction in photosynthetic activity might be due to photoinhibition due to strong irradiance in the central hours of the day [70].

Overall, the impact of freshening was less marked than that of salinization. During the freshening experiment, the first level of treatment mainly followed the same evolution pattern as the control, and strong changes were found at the lowest salinities only. On the contrary, for the salinization experiment, despite the choice of increasing salinity progressively through two successive additions of salt, effects already started to emerge after the first salinization, 24 h after the beginning (8 h for photosynthetic activity). This may indicate salinization effects can be more marked than decreased salinity ones; however, the salinity intervals established for the two experiments are not really comparable and must be interpreted keeping in mind the natural conditions. For the freshening experiment, the maximum salinity gap was 50% and salinity values changed from around 40 to around 20. This range fits the most frequent range of salinity found in the natural environment (Santa Giulia lagoon) during autumn, so it falls into the natural variability range. Nevertheless, the short-term application resulted in an abrupt decrease, which can explain the observed effects, and in fact mimic a flash flood event, as required by our starting point objective. On the other hand, for the salinization experiment, we applied a larger salinity span, moving from 43 to up to 73 in a very abrupt time interval, thus applying a strong osmotic stress. Such high values are the result of several days' evaporation in the field and were observed only sporadically in situ. In the future, we may expect salinization processes through evaporation to be more frequent, longer lasting, and more rapid, but the abruptness in salinity change, as well as intensity of the gap applied, can explain the major impact under our experimental conditions. Stefanidou et al. [22] found an opposite trend, meaning a major strong negative impact of decreased salinity compared to salinization on phytoplankton communities. Discrepancies with our findings might reside in the fact that the authors tested marine communities and small salinity ranges ($\pm 5$ PSU): the marine environment is a stable system compared to a lagoon, and marine communities are expected to be less tolerant towards disturbance than brackish ones, which are particularly tolerant towards low salinity conditions and larger salinity ranges [20]. Moreover, Mediterranean coastal lagoons are usually strongly dependent on watershed inputs and, as mentioned before, salinity ranges applied during the freshening experiment frequently occur in the natural environment from which the starting communities were collected for this study.

Overall, total growth inhibition observed under increasing salinity is not universally confirmed by other studies. Some previous works indicate that high salinity boosted the biomass and growth of phytoplankton communities [23,26,31,32,71]. This was most often due to a decrease in zooplankton grazing pressure and the application of salinity ranges on communities from different freshwater or brackish environments [26,71]. Some other studies in natural transitional environments actually corroborated our observations, highlighting an inhibition of biomass under increasing salinity [27,72]. A strong difference among treatments was observed in our study, indicating that diatoms, dinoflagellates, and chlorophytes were lost or experienced lower growth under increasing salinity stress. Alloxanthin and zeaxanthin variations also indicated a negative effect of increased salinity on cryptophytes and cyanobacteria. Overall, the negative effect of the treatment can be traced back to osmotic stress on the cells, as was already proposed by different studies on primary producers [29,73]. Microscopy observations at the end of the salinization experiment identified in particular the complete loss of some chlorophyte species and this is in accordance with other studies, like Redden and Rukminasari [72], who observed a reduction in chlorophytes in a coastal lake (Myall Lakes, NSW, Australia) under an increase in salinity, even if slight (+4 ppt, from 4 to 8 ppt). Moreover, chlorophytes and cryptophytes are generally associated with freshwater inputs in transitional systems, hence showing a general preference for low salinities [51,74–76]. For the chlorophytes, there was a contrast in fluorometry, pigment, and microscopy analyses, probably linked to the weakness of the BBE precision and pigment redundancy [51,63]. Nevertheless, the use of different methods was really helpful in constructing a more precise idea of community changes at different

levels. As for the loss of cyanobacteria under salinization, the results matched the cytometry analysis, identifying a strong loss of picocyanobacterial, particularly of the Synechococcus-like type. Initially dominating the small fraction of the phytoplankton community, at the end of the experiment picocyanobacteria had decreased in favor of picoeukaryotes and nanophytoplankton. Synechococcus is the dominant picophytoplankton taxon in rich coastal waters, which can explain the absence or paucity of Prochlorococcus-like detection over both experiments, as also observed in Ghar El Mehl lagoon (Tunisia) [77]. Synechococcus dynamics are strongly influenced by salinity [78]. In an estuarine context, it has already been proven that the community composition of Synechococcus populations varies according to salinity, by developing mostly marine, mostly freshwater lineages or mixed populations, allowing a wide range of salinity tolerance [78–80]. The loss of Synechococcus-like organisms under salinization might be due to a sudden strong osmotic pressure, which could not allow the establishment of a new population's composition, and maybe competition processes [81]. Competition control is also probably at the base of heterotrophic bacteria loss: all salinization treatments showed a decrease in bacteria contribution, especially in the control and lowest salinization level, where micro- and nanophytoplankton growth was higher. Top-down control through grazing and bottom-up control through the competition of phytoplankton with bacteria populations have already been described in mesocosm experiments [49].

Parallel to all these modifications, a strong negative effect of increased salinity on the photosynthetic activity of the community was detected, indicating a stress induced by salinization, affecting the community health. Supporting in general Fv/Fm values, the $\Phi$PSII values in salinization treatments, always lower than 0.3, indicate a low physiological state of the community [40]. Salt-stress-driven impacts on photosynthetic metabolism were already observed in marine and brackish communities or species, despite some variability in responses [82,83]. Le Rouzic [82] was able to detect Fv/Fm depletion induced by short-term salt stress in microalgal communities from salt marshes, but responses could not be directly linked to this stressor alone and strongly varied according to the dominant species present. Li et al. [84] highlighted the negative effects of acute stress salinity on estuarine phytoplankton communities, but the effect was short term (several tens of minutes), while, in contrast, we found a strong treatment effect in our study over the entire duration of the experiment, i.e., several days. The same author identified similar responses for a decreasing salinity gradient. D'ors et al. [28] demonstrated the strong effect of freshening on phytoplankton metabolic activities. Conversely, in our study, no negative effect on photosynthetic activity was found under decreasing salinity treatment. For this experiment, only treatment effects were identified for the community structure and composition. Globally, the lowest treatment did not differ much from control, showing lower picoeukaryote, picocyanobacteria, and nanophytoplankton abundances compared with stronger treatments towards the end of the experiment. In general, diatoms and chlorophytes were favored by freshening, together with prasinophytes, cyanobacteria, and cryptophytes. This is not surprising, as prasinophytes, chlorophytes, and cryptophytes in particular have been strongly associated with decreased salinization, both in experimental conditions and in the field, linked to freshwater inputs from the watershed and flash flooding events [34,51,76,85]. Total microphytoplankton abundance increased with decreased salinity, which is in contrast with similar studies, which generally identify a loss in biomass linked to lower salinity [30,32,33].

Most dinoflagellates disappeared under freshening treatments. This is not surprising, as most of dinoflagellates, like Prorocentrum micans, are generally associated with salinity increases [86]. However, the observed loss of Alexandrium sp. is more unexpected, as this taxon has already been linked to decreasing salinity processes in coastal environment [87]. This might be due to the emergence of an initially rare species, Akashiwo sanguinea, which probably took advantage of the low salinity conditions, which are in its growth optimum [88,89], and took the upper hand over other taxa. The initially rare diatom Entomoneis sp. also developed massively under freshening treatment, moving from initially

rare to dominant at the end of the experiment, to the detriment of other taxa. This genus is generally brackish and low-salinity tolerant [82,90,91]. Its development contributed to another important effect induced by freshening, i.e., the loss of diversity in the community. A similar tendency was also found for the highest salinization treatment despite not being significant. In both cases, these observations are compliant with other studies, indicating a loss of diversity in the short term under increased or decreased freshening stress on phytoplankton communities from various environments [22,29,92]. For instance, Flöder et al. [29] found that increased salinity in a tidally influenced lake (New Zealand) community induced the depletion of dominant phytoplankton species, which were replaced by the compensatory growth of initially rare ones. Barnes and Wurtsbaugh [26] highlighted a loss of diversity even at a higher trophic level under increased salinity in a saline lake, thus indicating a biodiversity loss under this kind of disturbance.

Overall, our results highlight changes in phytoplankton community structure and functioning which are compliant with other observations, even though there are sometimes contrasts with previous studies [22,23,26,29–32]. This could be due to differences in the starting communities and/or environments chosen [31–33]. Our results also highlighted strong impacts on the biomass, structure, and metabolism of the phytoplankton communities, and for both increased and decreased salinity. This work hence provides new knowledge regarding strong and sudden salinity fluctuations in lagoon environments, which might be useful for future research and comparison with other similar systems. The short-term changes identified indicate that potential transition processes towards completely new community configurations could occur in the middle to long term under salinity perturbation in the future.

### 4.2. Perspectives and Implications on Future Management of Small Mediterranean Lagoons

Numerous studies have proven that salinity changes in various environments, like coastal marine waters, saline or tidal lakes, or estuaries and lagoons outside the Mediterranean region, can induce modifications in biotic communities in terms of biomass, species composition and richness, diversity, and resource-use efficiency [22–24,26,29,33]. We demonstrated that such responses also took place in a Mediterranean coastal lagoon environment.

Projections for climate change identify Mediterranean coastal lagoons as climate-change hotspots. Predicted changes in these systems are expected to be amplified compared to perturbations in the open sea [14]. In particular, water temperature changes are predicted to be 15% greater than in the sea, parallel to salinity increases over forty times stronger than sea [14]. These effects are predicted to be especially evident in shallow lagoons, with a low connection to the sea, since they are more reactive and present a lower mitigation potential towards disturbances, making them excellent sentinels for global change [14,15]. Our study is hence a first fundamental step towards the anticipation and understanding of the destiny of these systems under future disturbances. We have already highlighted that salinity variations can influence the relationship between phyto- and bacterioplankton dynamics. We suggest that future studies should focus on the testing of more complex biotic communities, including macrophytes, zooplankton, and benthic invertebrates. For instance, Barnes and Wurtsbaugh [26] performed an experiment on salinity variation's effects on complex communities from the Great Salt Lake (UT, USA), which displays a wide salinity range. This made it possible to detect and describe complex trophic relationships and the combined effects of the environmental variations together with grazing and competition under salinity stress, thus giving a simulation of the potential effects of disturbance on communities as close as possible to natural conditions [26].

Similarly, it would be interesting to perform further investigations on the effects of salinity variations together with other factors or stressors. For instance, the potential of the synergic effects of salinity and nutrient variations on biotic communities has already been established [49,93]. In the natural environment, together with freshwater input, nutrient inputs from the watershed are also usually associated with rainfall events [49,94]. Moreover, together with water, new communities of freshwater affinity are added during these events;

it has been proven that these phenomena will take place during flash floods too, and hence they will become more frequent under climate change as it progresses [34,49]. Since we have already highlighted alterations in communities under decreased salinity, it would be interesting to further investigate responses coupled to reactions to the new communities' composition. For instance, Fouilland et al. [49] described the increase in Chlorophyceae, picoeukaryotes, and bacterioplankton from river-water inputs and the consequent effects on phytoplankton community dynamics, alongside responses linked to salinity variations.

Furthermore, an increase in temperature is also predicted under climate change progression. Other studies highlighted the synergistic effects of salinity variations and temperature increase on phytoplankton communities [22,33]. However, the effects were different: Stefanidou et al. [22] found a cumulative negative effect of heat shock coupled with increased or decreased salinity, while Hernando et al. [33] pointed out that increased temperature could compensate lipid damage under decreased salinity, thus mitigating its negative effects on biomass loss. Since we found that the responses of phytoplankton communities to changes in salinity were similar to those obtained in these studies, like alterations in community composition, biomass, and metabolism, it is likely that the community would react synergistically to the simultaneous presence of these two stressors.

Of course, these responses highly depend on the type of communities considered (marine, brackish, or freshwater), which makes it difficult to compare between experiments, as highlighted by several studies [31–33]. Moreover, the final composition of the communities strongly depends on the starting communities [31–33]. This must be considered to fully understand future responses, as they will depend on the communities present at the time of perturbation [33]. Since future projections identify an increase in potentially harmful species linked to climate change [80,94–97], and since we demonstrated that some potentially harmful species can emerge under salinity variations despite being initially rare, as was the case for the dinoflagellate Akashiwo sanguinea in the freshening experiment, we can expect a potential exacerbation of the occurrence and intensity of harmful algal blooms in the future. Many harmful and invasive species are highly tolerant to disturbance and display wide salinity tolerance ranges [89,98–100]. Since we demonstrated salinity variations can change the community composition through the compensatory growth of some tolerant species and the emergence of initially rare ones, we can expect these harmful taxa to appear more frequently and further contribute to biodiversity depletion and ecosystem degradation. Further specific research on the development of rare and harmful lagoon phytoplankton species under salinity stress induced by climate change could be an interesting step to better understand the potential consequences of their presence and development under future scenarios.

Changes in the size, structure, and composition of phytoplankton communities, especially the simplification and loss of biodiversity demonstrated during our study, can have negative impacts on higher trophic levels, as different phytoplankton groups present different nutritional values, especially regarding lipid content [92]. Changes at the phytoplankton level may also impact the capacity of higher trophic groups to face disturbance, due to, for instance, an insufficient lipid supply from phytoplankton reducing the resistance capacity of zooplankton towards salinity stress [92]. Hence, structural modifications of phytoplankton and the diversity loss identified under salinity stress might impact ecosystem functioning, contribute to the degradation of the resistance capacity of the system by exacerbating disturbance effects at multiple levels, and ultimately impact the ecosystem services provided [92,101].

In particular, hypersaline conditions have already been associated with the favoring of stress-tolerant and harmful species [31]. The Santa Giulia lagoon, the site chosen for this study, has already experienced a drying-up event, reaching salinity values up to 126, associated with long-lasting isolation from the sea; these extreme conditions, which have an increased probability of occurring under climate change, should be avoided in order to limit detrimental effects on biotic communities. In the instable transitional systems known as ICOLLs, communication with the sea is essential to regulate salt balance and the system

functioning [27,102]. The loss of connectivity of lagoons and other transitional systems to marine environment has already been associated with alterations in salinity and biotic communities [94,103,104]. The artificial opening of sea inlets is often used to control water discharges and sea exchanges in transitional systems [27]. With the progression of climate change, the natural opening of sea inlets will become more difficult due to an increase in evaporation and a decrease in the frequency and shortening of rainy periods, which normally ensure outgoing pressure and sediment flushing in sea inlets [101]. Moreover, despite climate change being a major driver for long-term environmental change, local environmental management is also strongly involved, as it was demonstrated by Fichez et al. [105]: the authors highlighted strong salinity variations due to freshwater-river-discharge increases in a tropical coastal lagoon (Laguna de Terminos, Mexico). This was independent of the consequences of climate change, and was rather due to deforestation in the watershed to build up space for agricultural fields. The results of this study demonstrate that management actions and decisions, even those involving the lagoon's surroundings, have power over and consequences for system functioning and its destiny, since they also are drivers, together with climate change, of modifications on a long-term scale.

Based on our results on potential responses to salinity variations together with these observations, we suggest the future conservation of small Mediterranean lagoons should be subordinated to the application of an integrative management approach [106–108]. This should focus on maintaining the hydrological continuum of these small lagoons to adjacent systems, through the management of activities in their watersheds and eventually artificial management of their sea channels, in order to promote a good circulation and avoid confinement. This would hopefully ensure that they avoid falling abruptly into these "extremes" and allow the control or mitigation of salinity variations induced by climate change.

**Author Contributions:** Conceptualization, V.L., M.G., N.M., P.C. and V.P.; methodology, V.L., M.G., N.M., L.A., R.B., A.A., P.C. and V.P.; validation, M.G., N.M., P.C. and V.P.; formal analysis, V.L.; investigation, V.L., M.G., N.M., L.S. and V.P.; resources, L.A., R.B. and A.A.; writing—original draft preparation, V.L.; writing—review and editing, V.L., M.G., N.M., P.C. and V.P.; visualization, V.L., M.G., N.M., P.C. and V.P.; supervision, M.G., N.M., P.C. and V.P.; project administration, V.P.; funding acquisition, V.P. All authors have read and agreed to the published version of the manuscript.

**Funding:** Viviana Ligorini was awarded a grant from the Corsican Regional Council and the University of Corsica. This study was supported by funding from the French Government and from the Corsican Regional Council (CPER Gerhyco project).

**Data Availability Statement:** The datasets generated and/or analyzed during the current study are available from the corresponding author on reasonable request.

**Acknowledgments:** The authors are grateful to the UAR 3514 Stella Mare (Université de Corse–CNRS) zoo- and phytoplankton and maintenance teams for their cooperation during experimental setup and coordination.

**Conflicts of Interest:** The authors declare no conflict of interest.

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
