# Peer review of "Response of Phytoplankton Communities to Variation in Salinity in a Small Mediterranean Coastal Lagoon: Future Management and Foreseen Climate Change Consequences"

_water, doi:10.3390/w15183214_

Round 1
Reviewer 1 Report
Review for the paper "Response of phytoplankton communities to variation in salinity in a small Mediterranean coastal lagoon: future management and foresee climate change consequences" by Viviana Ligorini, Marie Garrido, Nathalie Malet, Louise Simon, Loriane Alonso, Romain Bastien, Antoine Aiello, Philippe Cecchi, Vanina Pasqualini submitted to "Water".
Marine phytoplankton constitute a diverse group of pelagic, photosynthetic microorganisms, accountable for over 90% of oceanic primary production. These individual cells, whose sizes span four orders of magnitude, are found distributed worldwide. Remarkably, the global standing stock of phytoplankton undergoes complete turnover every 2 to 6 days on average, often resulting in the depletion of available nutrient resources.
Over a century of scientific study has illuminated the pivotal role played by marine phytoplankton in shaping the structure and overall functioning of marine ecosystems. Indeed, their influence stretches to domain as varied as fishery stocks, biogeochemical cycles, climate regulation, and weather patterns. The environmental pressures exerted on phytoplankton populations constitute one of the most formidable challenges currently being tackled within scientific research.
The paper at hand directs its focus towards phytoplankton communities populating select Mediterranean lagoons. It aimed to investigate the impact of salinity gradients in the formation and evolution of phytoplankton communities. To achieve this, the authors embarked on a series of experimental tasks and observed a detrimental impact of high saline conditions on the phytoplankton, although only minor influence was noted under conditions of low salinity.
The results of this study may prove applicable considering the significant climatic shifts observed over recent decades.
The manuscript is skillfully composed, and I was unable to identify any discernible scientific inaccuracies.
My only suggestion for authors is to add an explanation for abbreviations used in the table and figure captions.
Minor
Author Response
Please see the attachment, from page 2.

Reviewer 2 Report
Evaluation of the manuscript titled “Response of phytoplankton communities to variation in salinity in a small Mediterranean coastal lagoon: future management and foresee climate change consequences”.
The manuscript (ms.) evaluated deals with a series of experiments changing salinity to observe their effect upon phytoplankton in a coastal lagoon in the Mediterranean Sea. The manuscript is well-focused and well-written, and the amount or work, data acquired, conclusions and possible consequences allow to consider the manuscript (ms.) further for publication, but some minor to major questions and points should be considered.
Some questions and hypotheses are pretty obvious, and their context is predictable and expected: there should be an effect of the microalgae communities by changing salinity, especially by increasing salinity. However, the salinity changes proposed in the experiments seem to be too abrupt and are not comparable with “natural” conditions (this was already mentioned by the authors in line 580, in the Discussion). It is difficult to compare these experiments with “natural conditions” that may happen in the near future.
The flora found in shallow lagoons does not necessarily correspond to “microplankton” as part of the phytoplankton, but the flora may be a combination of various habitats and growth forms, for example, benthic and littoral microalgae. This is observed in the flora composition, where many genera reported in this study belong to the benthic community rather than the microplankton. The level of identification is poor with taxa not identified or identified only to family or to genus level (except for a couple of species identifications).
What are (were) the original environmental conditions and their seasonal fluctuations in the Santa Giulia lagoon? They are clearly indicated, and this is important to compare with the artificially created microcosmos and is especially interesting to consider changes in salinity.
Identification at “lowest” possible taxonomic level (line 293). This is wrong, as “species” is the “highest” taxonomic level. Please consider this.
The heterotrophic bacteria are not part of the phytoplankton.
Define ultra- and picoplankton: authors are using both terms indistinctly.
-----------------------------------------------------------------------------------------------------
My recommendation is that the current ms. should be published, but once the mentioned questions and points may be revised.
Author Response
Please see the attachment, from page 3.

Reviewer 3 Report
The paper designed the experiments to assess the acute effects of salinization and flood events on phytoplankton communities in a Mediterranean lagoon. The experiments were well designed, and the results were well presented.
Although I wish I was able to see more about how rare species (especially the harmful ones) could be developed during the processes, which means more experiments, marine science always needs more studies and data. Regardless, the paper can be published in the current format by my standard.
Author Response
Please see the attachment, from page 6.
